# DIFFUSION TRANSFORMER POLICY

## ABSTRACT

Recent large visual-language action models pretrained on diverse robot datasets have demonstrated the potential for generalizing to new environments with a few in-domain data. However, those approaches usually predict discretized or continuous actions by a small action head, which limits the ability in handling diverse action spaces. In contrast, we model the continuous action with a large multimodal diffusion transformer, dubbed as Diffusion Transformer Policy, in which we directly denoise action chunks by a large transformer model rather than a small action head. By leveraging the scaling capability of transformers, the proposed approach can effectively model continuous end-effector actions across large diverse robot datasets, and achieve better generalization performance. Extensive experiments demonstrate Diffusion Transformer Policy pretrained on diverse robot data can generalize to different embodiments, including simulation environments like Maniskill2 and Calvin, as well as the real-world Franka arm. Specifically, without bells and whistles, the proposed approach achieves state-of-the-art performance in the Calvin novel task setting, and the pretraining stage significantly facilitates the success sequence length on the Calvin by over 1.2. The code will be publicly available.

## 1 INTRODUCTION

Traditional robot learning paradigm usually relies on large-scale data collected for a specific robot and task, but collecting robot data for generalist tasks is time-consuming and expensive due to the limitations of robot hardware in the real world. Nowadays, the foundational models OpenAI (2022; 2023; 2021); Rombach et al. (2021) in Natural Language Process and Computer Vision, pretrained on broad, diverse, task-agnostic datasets, have demonstrated powerful ability in solving downstream tasks either zero-shot or with a few task-specific samples. It is principally possible that a general robot policy exposed to large scale diverse robot datasets improves generalization and performance on downstream tasks Brohan et al. (2022; 2023). However, it is challenging to train a general robot policy on a large scale of cross-embodiment datasets with diverse sensors, action spaces, tasks, camera views, and environments.

Toward a unified robot policy, existing works directly map visual observation and language instructions to actions with large visual-language-action models for robot navigation Shah et al. (2023a;b) or manipulation Brohan et al. (2022; 2023); Kim et al. (2024); Team et al. (2024), and demonstrate zero-shot or few-shot generation to new environments. Robot Transformers Brohan et al. (2022; 2023); Padalkar et al. (2023) present robot policy based on transformer architecture, and demonstrate robust generalization by training on the large scale of Open X-Embodiment Dataset Padalkar et al. (2023). Octo Team et al. (2024) follows the autoregressive transformer architecture with a diffusion action head, while OpenVLA Kim et al. (2024) discretizes the action space and leverage the pretrained visual-language model to build VLA model exposed to Open X-Embodiment Dataset Padalkar et al. (2023). Though those Visual-Language-Action (VLA) models Team et al. (2024); Kim et al. (2024) have shown the potential to learn robot policy from the large cross embodiment datasets Padalkar et al. (2023), the diversity of robot space among the cross embodiment datasets still limits the generalization of previous models with autoregressive transformers.

Recent diffusion policy Chi et al. (2023); Ze et al. (2024); Ke et al. (2024) has shown its stable ability in robot policy learning for single task immitation learning with UNet or cross attention architecture, and diffusion transformer demonstrates its scalability in multi-modal image generation Peebles & Xie (2023). Specifically, Octo Team et al. (2024) presents a generalist policy that denoises the

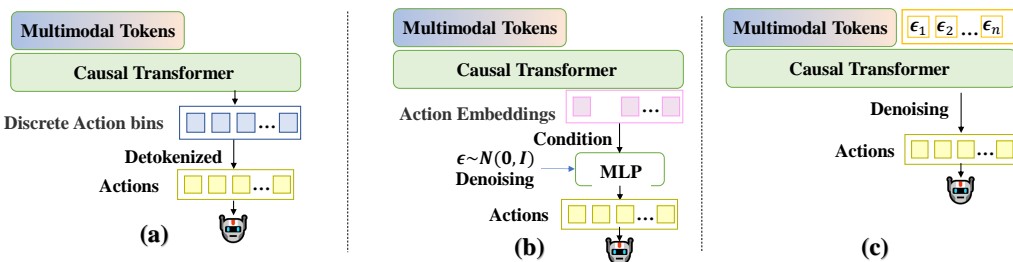

Figure 1: Illustrations of different robot policy architectures. (a) is the common robot transformer architecture with discretization actions, *e.g.*, Robot Transformer Brohan et al. (2022; 2023) and OpenVLA Kim et al. (2024). (b) is the transformer architecture with diffusion action head which denoises the continuous action with a small MLP with embedding from the causal transformer, *e.g.*, Octo Team et al. (2024). (c) is the proposed Diffusion Transformer architecture that utilizes the large transformer to denoise actions in an in-context conditioning style.

action with a small MLP network conditioned on a single embedding of auto-regressive multi-modal transformer. However, the robot space of large-scale cross-embodiment datasets contains various cameras views and diverse action spaces, which poses a significant challenge for a small MLP to denoise the continuous action conditioned on a single action head embedding.

In this paper, we design a Diffusion Transformer architecture for generalist robot policy learning. Similar to previous robot transformer models Brohan et al. (2022; 2023); Padalkar et al. (2023); Team et al. (2024); Kim et al. (2024), we leverage the transformer as our base module to retain the scalability on the large-scale cross-embodiment datasets. Different from Brohan et al. (2022; 2023); Padalkar et al. (2023); Team et al. (2024); Kim et al. (2024), we present an in-context conditional diffusion transformer architecture to denoise the action chunks, rather than utilizing a small MLP to denoise action embedding to continuous actions as illustrated in Figure 1. The Diffusion Transformer Policy retains the scalability of transformer for diffusion, and thus more effectively generalizes action policy from the large diverse datasets.

In a nutshell, we present a Diffusion Transformer Policy, that incorporates a causal transformer as an in-context conditional diffusion backbone and denoise continuous action chunks with the transformers. Extensive experiments demonstrate Diffusion Transformer Policy achieves considerably better performance on two large-scale Sim datasets, Maniskill2 and Calvin, compared to corresponding diffusion action head Team et al. (2024) and discretized action head Brohan et al. (2022) baselines. Meanwhile, The proposed model trained on the Open X-Embodiment Dataset achieves better generalization performance compared to the baseline methods on the Real Franka platform.

## 2 RELATED WORK

**Diffusion Policy** Denoise diffusion techniques Ho et al. (2020); Rombach et al. (2022) are pioneering image generation, and recent Diffusion Policy Chi et al. (2023); Ze et al. (2024); Ke et al. (2024) has exhibited a powerful ability in modeling multimodal actions compared to previous robot policy strategies in both 2D Chi et al. (2023) and 3D observations Ze et al. (2024); Ke et al. (2024). Current diffusion policy approaches usually follow an Unet structure or a shallow cross-attention network for a single manipulation task, leaving large-scale multimodal diffusion policy poorly investigated. For example, 3D diffusion Policy Ze et al. (2024) presents a diffusion approach conditional on a 3D point cloud, while 3D diffuser actor Ke et al. (2024) proposes a 3D diffusion strategy based on point cloud with cross-attention. Differently, we present a scalable in-context conditioning diffusion transformer architecture for generalist robot policy. Recent generalist policy Octo Team et al. (2024) conditions the denoise process on the embedding from the Transformer model with a small MLP diffuser. By contrast, the diffuser in Diffusion Transformer Policy is a large Transformer architecture.

**Generalist Robot Policies** The embodiment community has shown increasing interest in generalist robot policy with foundational multi-modal models for both robot navigation Shah et al. (2023a;b);

Yang et al. (2024); Sridhar et al. (2024); Huang et al. (2023) and manipulation Bousmalis et al. (2023); Brohan et al. (2022); Shah et al. (2023c); Shridhar et al. (2023); Brohan et al. (2023); Padalkar et al. (2023); Kim et al. (2024); Team et al. (2024). Recent approaches Brohan et al. (2022; 2023); Padalkar et al. (2023); Kim et al. (2024); Team et al. (2024) aim to achieve generalist policy with scalable Visual-Language-Action models. We follow this paradigm to approach generalist and adaptive robot policy. Brohan et al. (2022; 2023); Padalkar et al. (2023); Kim et al. (2024) construct the action token by discretizing each dimension of the robot actions separately into 256 bins. However, this discretization strategy incurs internal deviation in robot execution. Unlike those methods, we present a Diffusion Transformer Generalist Policy, which denoises the continuous actions with a large Transformer model. The proposed approach retains the scalability of the Transformer and meanwhile facilitates the modeling of cross-embodiment action chunk representations. Meanwhile, the Diffusion Transformer Policy aligns robot action together with the language instructions and image observations as an in-context conditional style.

## 3 METHOD

We describe the proposed architecture of diffusion transformer policy in this section, a DiT-based generalist diffusion policy model that can be adapted to new environments and embodiment. The model, built from a diffusion transformer architecture, achieves better generalization on both novel camera views and environments from large amounts of diverse robot data.

### 3.1 ARCHITECTURE

**Instruction Tokenization**. The language instructions are tokenized by a frozen CLIP Radford et al. (2021) model.

**Image observation Tokenization**. The image observations first pass into the DINOv2 Oquab et al. (2023) to obtain the image tokens. Note that DINOv2 Oquab et al. (2023) is trained on the web data which is different from the robot data, we thus jointly optimize the DINOv2 parameters together with Transformers through an end-to-end way.

**Q-Former**. To reduce the computation cost, a Q-Former together with FiLM Perez et al. (2018) conditioning is incorporated to select image tokens from the features of DIDOv2 Oquab et al. (2023) by instruction context.

**Action Tokenization**. We use the end-effector action and represent each action with a 7D vector, including 3 dimensions for translation vector, 3 dimensions for rotation vector, and a dimension for gripper position. To align the dimension with image and language tokens, we simply pad the action vector with zeros to construct the action token. We only add the noise into the 7D action vector during denoise diffusion optimization.

**Architecture**. Our core design is the Diffusion Transformer structure Peebles & Xie (2023) which denoises action token chunks, instead of each single action token, conditioned on image observation and instruction tokens by an in-context conditioning style with a causal transformer network, *i.e.*, we simply concatenate image tokens, language tokens, and timestep token in the front of the sequence, equally treating the noisy action tokens from the image/instruction tokens as illustrated in Figure 2. This design retains the scaling properties of transformer networks. The model, conditioned on language instructions and image observations with the causal transformer structure, is supervised by the noise that we add to the continuous actions. In other words, we conduct the diffusion objective directly in the action chunk space with a large transformer model, differently from a diffusion action head with a few MLP layers Team et al. (2024).

The proposed Diffusion Transformer Policy is a general design that can be scaled to different datasets, and demonstrate excellent performance. Meanwhile, we can also add additional observation tokens and input into the transformer structure. Appendix A provides more details.

### 3.2 TRAINING OBJECTIVE

In our architecture, the denoising network $\epsilon_\theta(\boldsymbol{x}^t, c_{obs}, c_{instruction}, t)$ is the whole causal transformer, where $c_{obs}$ is the image observation, $c_{instruction}$ is the language instruction, and $t \in$

$1, 2, ...T$ is step index in our experiments. During the training stage, we sample a Gaussian noise vector $x^t \in \mathcal{N}(0, I)$ at timestep $t$, where $T$ is the number of denoising timesteps, and add it to action $a$ as $\hat{a}$ to construct the noised action token, finally predicting the noise vector $\hat{x}$ based on the denoising network $\epsilon_\theta(\hat{a}, c_{obs}, c_{instruction}, t)$, where $t$ is randomly sampled during training. We optimize the network with MSE loss between $x^t$ and $\hat{x^t}$.

To generate an action, we apply $T$ steps of denoising with the optimized transformer architecture $\epsilon_\theta$ from a sampled gaussian noise vector $x^T$ as follows,

$$x^{t-1} = \alpha(x^t - \gamma\epsilon_\theta(x^t, c_{obs}, c_{instruction}, t) + \mathcal{N}(0, \sigma^2 I)).$$

where $\alpha$, $\gamma$, $\sigma$ is the noise scheduler Ho et al. (2020). In our experiments, $\epsilon_\theta$ is to predict the noise that adds to the action.

### 3.3 PreTraining Data

To evaluate the proposed Diffusion Transformer Policy, we choose Open X- Embodiment datasets Padalkar et al. (2023) for pretraining the model. We mainly follow Team et al. (2024); Kim et al. (2024) to choose the datasets and set the weights for each dataset. We normalize the actions similarity to Padalkar et al. (2023) and filter out outlier actions in the dataset. Additional details are provided in Appendix B.

### 3.4 Pretraining Details

We devise the proposed Diffusion Transformer architecture and evaluate the pretraining approach in the large cross-embodiment datasets Padalkar et al. (2023). We use the DDPM Ho et al. (2020) diffusion objective in the pretraining stage with $T = 1000$ for the Open X-Embodiment dataset Padalkar et al. (2023), while we set $T = 100$ with DDIM Song et al. (2020) for zero-shot evaluation to accelerate the inference. According to the preliminary experiment from Maniskill2 Gu et al. (2023), we use 2 observation images and predict 32 action chunks. We filter out the action chunks that include outlier values. We train the network with AdamW Loshchilov (2017) by 100,000 steps. We set the learning rate of the casual transformer and Q-Former as 0.0001, the learning rate of the pretrained image tokenizers as 0.00001, and the batch size as 8902. More pretraining details are provided in the Appendix.

## 4 Experiments

We evaluate the proposed methods with two baselines in three environments. We leverage Maniskill2 to present the ability of Diffusion Transformer Policy on large scale novel view generalization. Meanwhile, we demonstrate the generalization of the pretrained Diffusion Transformer Policy on CALVIN benchmark. Lastly, we further show the generalization of Diffusion Transfromer Poicliy on Real Franka Arm.

### 4.1 Baselines

**Discretization Action Head** We implement the RT-1 Brohan et al. (2022) style baseline models with a similar structure as ours. We maintain the Instruction Tokenization and Image Tokenization strategy in the proposed method. Different from ours, we follow RT-1 Brohan et al. (2022) to discretize each dimension of the action into 256 bins, and leverage the transformer network to predict the action bin indexes. Following Brohan et al. (2022; 2023), we use cross-entropy loss to optimize the network.

**Diffusion Action Head**. We also implement a diffusion action head strategy Team et al. (2024) in our architecture. Specifically, we utilize a three-layer MLP network as our denoising network condition on the output of each action token embedding by the same transformer architecture as ours.

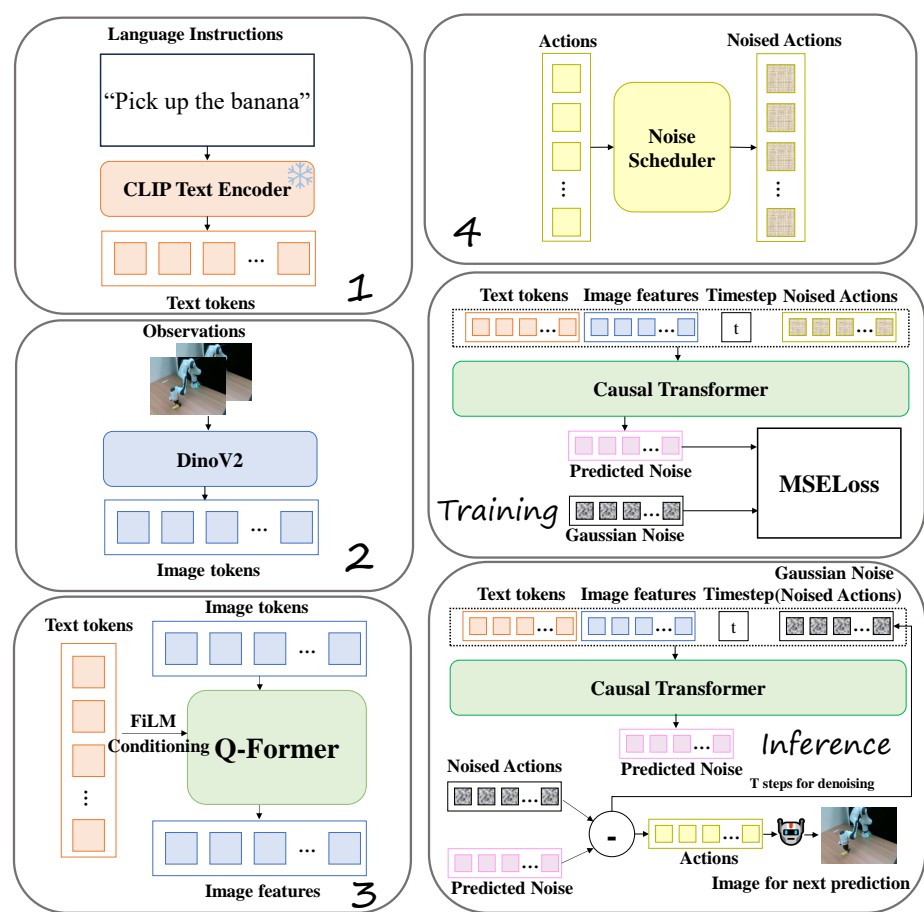

Figure 2: Our model is a Transformer diffusion structure. The model first incorporates a pretrained CLIP network to obtain instruction tokens. Meanwhile, we use the DINO-V2 Oquab et al. (2023) model to encode image observations, followed by a Q-Former to query observation tokens for each image observation. Next, we concatenate instruction tokens, image observation tokens, timestep token, and noised action tokens together to construct a token sequence as the input for transformer network to denoise the raw actions.

## 4.2 MANISKILL2

**Maniskill2** Gu et al. (2023) is the next generation of the SAPIEN Maniskill benchmark Mu et al. (2021), which is widely used to evaluate the generalized manipulation ability of the embodied models. It contains 20 different manipulation tasks families and over 4 million demonstration frames with different settings, including rigid/soft body, single/dual arm, etc. Maniskill2 also provides a fast and easy way to change the camera view and replay the trajectories. It is useful for the researchers working on generalized policy.

*Setup*. In our experiments, we select 5 tasks (PickCube-v0, StackCube-v0, PickSingleYCB-v0, PickClutterYCB-v0, PickSingleEGAD-v0) from Maniskill2, and then construct a camera pool with 300,000 random cameras, then sample 20 cameras from the camera pool to render a trajectory each time, and finally obtain about 40K trajectories totally. Moreover, we split the dataset into training set and validation set according to a ratio of 19:1. During the splitting, it is guaranteed that the single trajectory rendered under different camera views will appear in either training set or validation set in order to avoid data leaking. Specially, there are 74 different categories to pick and place in the task family PickSingleYCB-v0. In addition to the mentioned rules, we ensure each category can be found in both training set and validation set. After that, we sample 100 trajectories for each task family randomly from the validation set, constructing a close loop evaluation dataset with 500

Table 1: Comparision on Maniskill2 (success rate). SingleYCB indicates PickSingleYCB, ClutterYCB indicates PickClutterYCB, SingleEGAD indicates PickSingleEGAD. Disc ActionHead indicates Discretized Action Head strategy Brohan et al. (2022), while Diff ActionHead shows Diffusion Action Head Team et al. (2024).

| Method | All | PickCube | StackCube | SingleYCB | ClutterYCB | SingleEGAD |
|---|---|---|---|---|---|---|
| Disc ActionHead | 30.2% | 41.0% | 33.0% | 22.% | 1.0% | 54.0% |
| Diff ActionHead | 58.6% | **86.0%** | 76.0% | 37.0% | 24.0% | 70.0% |
| DiT Policy(ours) | **65.8%** | 79.0% | **80.0%** | **62.0%** | **36.0%** | **72.0%** |

trajectories in total. While training, considering the balance between different task families, we adjust the number of data pieces to the same by simply copying the trajectories from task family with fewer trajectories originally. The ability of the model is measured by the success rate of each task family while executing the close loop evaluation dataset.

*Optimization Details*. We optimize the network with AdamW Loshchilov (2017) by 50,000 steps on Maniskill2 and we set the learning rate as 0.0001. The number of training timesteps $T$ is 100 in Maniskill2 and the global batch size is 1024.

*Comparisons*. Table 1 compares the proposed method with discretized action head Brohan et al. (2022; 2023) and diffusion action head Team et al. (2024). Here, the proposed method use the same backbone and transformer architecture as the baseline methods. The experiments demonstrate Diffusion Transformer Policy achieves better results compared to Discretization Action Head strategy Brohan et al. (2022) under the large scale novel view maniskill2 benchmark. Specifically, we observe the proposed Diffusion Transformer Policy achieves clear better performance compared to the baselines. Meanwhile, Diffusion Transformer Policy demonstrates better performance in more complex tasks, *e.g.*, Diffusion Transformer Policy improves diffusion action head Team et al. (2024) by 20% in task PickSingleYCB and by 12% task PickClutterYCB. Those experiments show that Diffusion Transformer Policy achieves better scalability in the large scale diverse datasets, and meanwhile achieves better generation in camera view generalization.

## 4.3 CALVIN

CALVIN (Composing Actions from Language and Vision) Mees et al. (2022) is an open-source simulated benchmark to learn long-horizon language-conditioned tasks. CALVIN Mees et al. (2022) includes four different scenes tagged as ABCD and presents a novel scene evaluation benchmark, ABC→D, *i.e.*, trained on environments A, B, and C and evaluated on environment D. The goal of CALVIN is to solve up to 1000 unique sequence chains with 5 distinct subtasks. The benchmark requires successfully solving the task sequence with 5 continuous subtasks, and one of the important evaluation indicators is the success sequence length.

*Setup*. In this section, we utilize CALVIN (ABC→D) to evaluate the novel task generalization of Diffusion Transformer Policy architecture. Specifically, we directly apply the proposed method on CALVIN with a single static RGB camera and predict the end-effector action, including 3 dimensions for translation, 3 dimensions for Euler angles rotation and 1 dimension for gripper pose. We evaluate Diffusion Transformer Policy and Diffusion Action Head Team et al. (2024) on CALVIN, and leverage the pretrained model on Open X-Embodiment to initialize the model for CALVIN.

*Optimization Details*. While training Calvin, 2 history images are used as input. For each iteration, the model predicts 10 future frames supervised by MSE loss. An AdamW optimizer is used together with a decayed learning rate with half-cycle cosine scheduler after several steps of warming up. The learning rate is initialized as 1e-4. We use 4 NVIDIA A100 GPUs(80GB) to train the model for 15 epochs with a global batch size of 128.

*Comparisons*. Table 2 presents the comparisons with previous methods on Calvin and the proposed methods. Without whistles and bells, the proposed Diffusion Tranformer Policy achieves the state-of-the-art results. Particularly, we only use RGB camera stream for observation. The superior demonstrates the effectiveness of Diffusion Transformer Policy. Meanwhile, the pretraining on Open X-Embodiment Datasets significantly facilitates the performance by 1.23, which demonstrates the transferability of Diffusion Transformer Policy. By contrast, the performance of diffusion

Table 2: The comparison on Calvin Benchmark. 'S' indicates Static RGB, 'G' indicate Gripper RGB. 'SD' indicate Static RGB-D, 'GD' indicates Gripper RGB-D, 'P' indicates Proprio, *i.e.*, the observation arm position, 'C' indicates camera parameters.

| Method | Input | \multicolumn{6}{c}{No. Instructions in a Row (1000 chains)} | | | | | |
|---|---|---|---|---|---|---|---|
| | | 1 | 2 | 3 | 4 | 5 | Avg.Len. |
| SPIL Zhou et al. (2024) | S,G | 74.2% | 46.3% | 27.6% | 14.7% | 8.0% | 1.71 |
| RoboFlamingo Li et al. (2023) | S,G | 82.4% | 61.9% | 46.6% | 33.1% | 23.5% | 2.47 |
| SuSIE Black et al. (2023) | S | 87.0% | 69.0% | 49.0% | 38.0% | 26.0% | 2.69 |
| GR-1 Wu et al. (2023) | S,G,P | 85.4% | 71.2% | 59.6% | 49.7% | 40.1% | 3.06 |
| 3D Diffuser Ke et al. (2024) | SD,GD,P,C | 92.2% | 78.7% | 63.9% | 51.2% | 41.2% | 3.27 |
| diffusion head w/o pretrain | S | 75.5% | 44.8% | 25.0% | 15.0% | 7.5% | 1.68 |
| diffusion head | S | 94.3% | 77.5% | 62.0% | 48.3% | 34.0% | 3.16 |
| Ours w/o pretrain | S | 89.5% | 63.3% | 39.8% | 27.3% | 18.5% | 2.38 |
| Ours | S | **94.5%** | **82.5%** | **72.8%** | **61.3%** | **50.0%** | **3.61** |

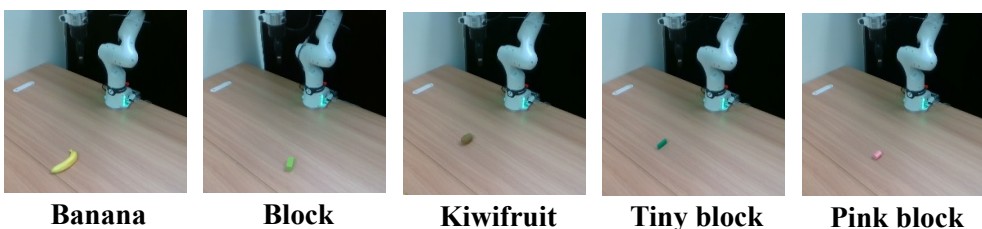

**Banana**      **Block**      **Kiwifruit**      **Tiny block**      **Pink block**

Figure 3: Illustration of Franka environment and corresponding objects.

action head is worse than Diffusion Transformer Policy by 0.45, though we load similar pretraining weights for diffusion head architecture. Diffusion Transformer Policy can scale across different environments, *e.g.*, transferring the knowledge from the diverse real datasets to the CALVIN dataset.

## 4.4 REAL FRANKA ARM

We finally evaluate the proposed method on the Real Franka Arm. Specifically, we train the model on Open X-Embodiment Datasets Padalkar et al. (2023), and evaluate it in our Franka Arm environment under zero-shot generalization and few-shot finetuning generalization.

*setup*. We set up the franka on the table with a black background. Meanwhile, we use a single third-person RGB camera about 1.5 meters away from the Franka Arm. Please refer to Figure 3 for the visualized demonstration. Considering the environment of our Franka setup is different from the scenes in Open X-Embodiment Stone et al. (2023), we mainly evaluate the proposed method on out-of-the-box generation and few-shot generation. In our experiments, we evaluate each model with the same scene, and the object is placed in 9 similar positions in a 9-grid format in front of the franka arm. Meanwhile, we maintain a small variance in those positions placing the objects for evaluation.

To evaluate the ability of the proposed model on new environments with a few demonstrations. We set 5 tasks, including "Pick up the green block", "Pick up the kiwifruit", "Pick up the banana", "Pick up the tiny green block", "Pick up the pink block". Meanwhile, we collect 50 trajectories for each task in the first three tasks, while leaving the remaining two tasks ("Pick up the tiny green block" and "Pick up the pink block") for out-of-distribution evaluation. Figure 3 presents the scenes and tasks. The image in Figure 3 is the exact image for our model. Our real-world environment is challenging since the object is small compared to the whole scene.

*Finetuning details.* In our experiments, we finetune the proposed method on the real Franka Arm with Lora Hu et al. (2021) and AdamW for 10,000 steps. We set the number of timesteps as 100 for DDPM Ho et al. (2020), and batch size as 512. Meanwhile, we finetune all the networks with one observation and one step prediction.

**Zero-shot Generalization** We directly take the models pretrained on Open X-Embodiment to evaluate zero-shot generalization in our environments. We compare the proposed method with the dis-

Table 3: Zero-Shot Comparision on Real-Franka. Here the baseline is the model with discretization actions, similar to RT-1 Brohan et al. (2022), that we implement and train on Open X-Embodiment.

| Method | Baseline | OpenVLA Kim et al. (2024) | Octo Team et al. (2024) | Ours |
|---|---|---|---|---|
| **PickBlock** | 0 | 0 | 0 | 10% |
| **PickBanana** | 0 | 0 | 0 | 15% |
| **PickKiwifruit** | 0 | 0 | 0 | 0 |

Table 4: Comparision on Real-Franka with few-shot fine-tuning. We represent the values in success rate (%). Task-1 is "Pick up the green block", Task-2 is "Pick up the banana", Task-3 is "Pick up the Kiwifruit", Task-4 is unseen object task "Pick up the green tiny block", Task-5 is unseen object task "Pick up the pink oval block".

| Method | All | Task-1 | Task-2 | Task-3 | Task-4 | Task-5 |
|---|---|---|---|---|---|---|
| Discretized Action Head Brohan et al. (2022) | 19.3% | 29.6% | 51.9% | 14.8% | 0 | 0 |
| Diffusion Action Head  Team et al. (2024) | 34.8 % | 40.7% | 85.2% | 25.9% | 22.2% | 0 |
| DiT Policy (ours) | **46.9%** | **55.6%** | **90.3%** | **44.4%** | **37.0%** | **7.4%** |

cretization action head baseline Brohan et al. (2022) (RT-1) that are optimized on the same dataset mixtures, as well as the released OpenVla Kim et al. (2024) and Octo-base Team et al. (2024) models. Besides, we evaluate it on three simple tasks: "pick up the green block", "pick up the banana", and "pick up the kiwifruit". Table 3 shows OpenVLA and Octo without in-domain finetuning fail to pick up the object, while the proposed method can pick the green block with a success rate of 10%. We think it is because the proposed model is able to scale better the mixture data from Open X-Embodiment. We provide a visualized analysis in Appendix C.1, where we find the OpenVLA and Octo fail to pick due to wrong grasp pose, while the proposed model can grasp the object with a small success rate.

**Few-shot Generalization** Table 4 presents the performance of the proposed Diffusion Transformer Policy compared to baseline methods. We observe different objects demonstrate various performances according to their attributes. The banana is the easiest object to pick up because the banana is longer, while kiwifruit is fat compared to other objects and all models achieve poor performance. The proposed Diffusion Transformer Policy effectively improves the diffusion action head according to Table 4. We find the discretized action head baseline achieves poor performance. Meanwhile, the Diffusion Transformer Policy is still able to pick up the novel object (*e.g.*, the pink object) with a low success rate, while the baseline methods totally fail.

### 4.5 ABLATION STUDY

In this section, we ablate some of the important designs of the model architecture, including the length of horizon, the length of observation, execution steps for evaluation on Maniskill2.

**Trajectory length.** The length of action chunks has an important effect on the performance of different tasks. Table 5 demonstrates that the performance increases as the increase of trajectory length. Meanwhile, we notice the performance of more complex tasks, *e.g.*, PickClutterYCB, significantly increases with the increase of trajectory length, while the easy task, *e.g.*, PickCube, maintains high performance after the trajectory length is more than 4. Meanwhile, the long horizon optimization significantly facilitates the performance since long horizon optimization is able to provide the target object position and help the model understand the localization of the object. For example, task PickClutterYCB with multiple YCB objects, requires the model to understand which one is the corresponding object.

**Observation length.** In our experiments, we find the length of history observation images also significantly affects the performance. At first, the performance significantly drops when we increase the length of observation history to 3. It might because it is more difficult for the model to converge with more observations since the number of corresponding image tokens also increases. Secondly, we observe using two image observations is more helpful for the performance when the prediction horizon is long. For example, when the length of trajectory is 32, the experiment with two observa-

Table 5: Ablation on Maniskill2. #obs indicates the number of history observation images. #traj shows the length of trajectory, *i.e.*, the sum length of observation and action prediction chunks. SingleYCB indicates PickSingleYCB, ClutterYCB indicates PickClutterYCB, SingleEGAD indicates PickSingleEGAD.

| #obs | #traj | All | PickCube | StackCube | SingleYCB | ClutterYCB | SingleEGAD |
|------|-------|-----|----------|-----------|-----------|------------|------------|
| 2 | 2 | 40.8% | 68.0% | 54.0% | 33.0% | 9.0% | 40.0% |
| 2 | 4 | 51.6% | 81.0% | 69.0% | 44.0% | 11.0% | 53.0% |
| 2 | 8 | 62.4% | **89.0** % | 78.0% | 54.0% | 25.0% | 66.0% |
| 2 | 16 | 65.6% | 83.0% | **80.0** % | **70.0** % | 25.0% | 70.0% |
| 2 | 32 | **65.8** % | 79.0% | **80.0** % | 62.0% | **36.0** % | **72.0**% |
| 1 | 32 | 61.6% | 78.0% | 76.0% | 64.0% | 24.0% | 66.0% |
| 1 | 1 | 51.0% | 79.0% | 66.0% | 42.0% | 19.0% | 49.0% |
| 3 | 3 | 35.4% | 54.0% | 49.0% | 27.0% | 5.0% | 42.0% |

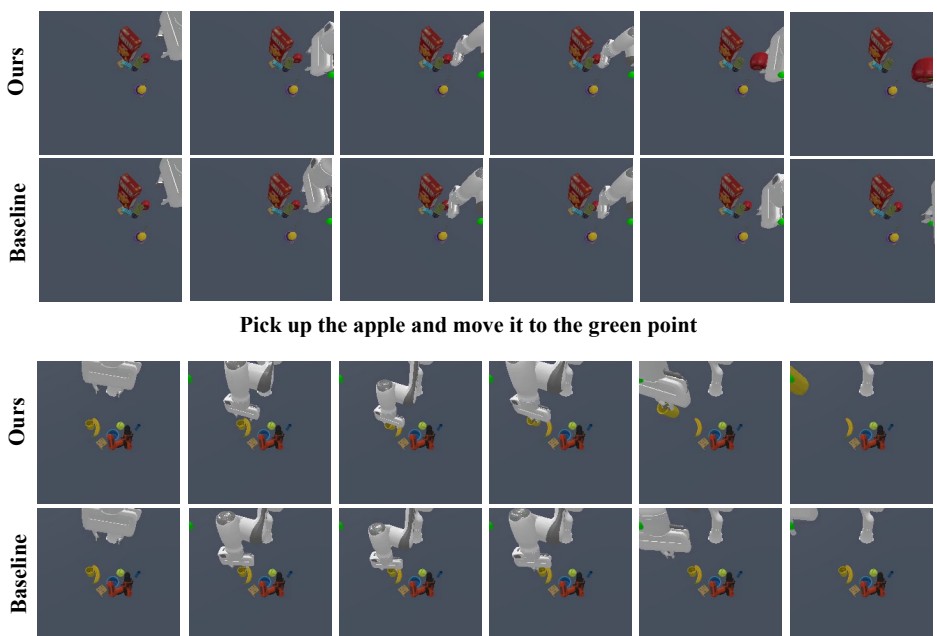

**Pick up the apple and move it to the green point**

**Pick up the cup and move it to the green point**

Figure 4: Visualized comparison between Diffusion Transformer Policy and Diffusion Action Head baseline on Maniskill2 (PickClutterYCB). The first raw is Diffusion Tranformer Policy, while the second raw is the baseline method with Diffusion Action Head.

tions achieves better performance. We think two observations can provide the visualized difference between two positions, and the difference of continunous gripper position indicates the action. The visualized difference is beneficial for future action prediction. However, for short horizon, the model majorly learns the projection from current observation to the corresponding actions.

**Execution steps.** Since the proposed model is able to predict multiple future actions, we can execute multiple steps in one inference. Here, we ablate the effect of execution steps using a model that has a trajectory length of 32 for quick evaluation in Table 6. The ablation study shows that the short execution steps are slightly better longer execution steps, *i.e.*, the farther away from the current frame, the worse the prediction quality.

### 4.6 VISUALIZED COMPARISON

**Maniskill2**. The proposed Diffusion Transformer Policy is able to model better action sequences. We conduct visualized analysis between the proposed method and the diffusion action head baseline

Table 6: The effect of the number of Execution steps on Maniskill2. #steps indicates the number of steps that we execute each prediction.

| #steps | 1 | 2 | 4 | 8 | 16 |
|---|---|---|---|---|---|
| All | 61.6% | 60.8 % | 60.6 % | 60.0 % | 58.0 % |

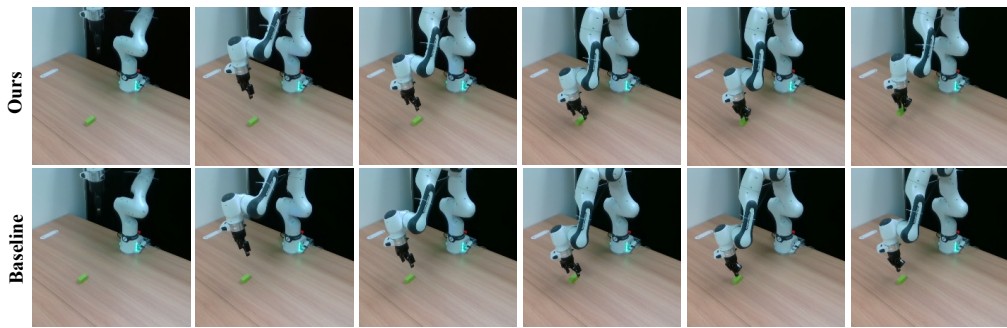

Figure 5: Visualized comparison between Diffusion Transformer Policy and Diffusion Action Head strategy on Real Franka Arm (Pick up the green block). The first raw is Diffusion Tranformer Policy, while the second raw is the baseline method with Diffusion Action Head.

on Maniskill2 in Figure 4. We select two trajectories from PickClutterYCB task, which is the most challenging task in Maniskill2. Figure 4 presents the grasp position is significantly important for picking up successfully, and the main reason that the baseline fails to pick up is the wrong grasp position. Meanwhile, we observe the major challenge of task PickClutterYCB is the grasping position prediction, especially when the target object is near by other objects. Compared to the diffusion action head baseline, Diffusion Transformer Policy is able to predict better action chunks for correctly picking the object with a suitable end-effector pose.

**Real Franka Arm**. We also illustrate the comparison between the Diffusion Transformer Policy and diffusion action head baseline on real Franka Arm in Figure 5. We demonstrate the experimental results under few-shot finetuning setting. We find the proposed method achieves better action prediction when the Gripper is approaching the object and finally picks up the small green object successfully, while the baseline fails to pick up due to the inaccurate grasp position. Meanwhile, we observe that failures are usually caused by a tiny position bias and we can not even directly discriminate the position by eyes from the image. For those cases, we argue the diffusion transformer policy has learnt better grasp position during the pretraining stage, and thus reduce failure rate due to the wrong grasp pose, while it is difficult for the diffusion action head. We demonstrates more failure cases to analyze the challenges in the Franka Arm.

## 5 CONCLUSION, DISCUSSION AND FUTURE WORK

In this paper, we present a Diffusion Transformer architecture for generalist robot learning, named as Diffusion Transformer Policy. Diffusion Transformer Policy directly utilizes the large transformers as a denoising network to denoise the continuous actions conditioned on language instruction and image observations. The proposed architecture retains the scale attribute of the transformer, thus is capable of generalizing to different datasets with a unified architecture. Extensive experiments on Maniskill2, CALVIN, real Frank Arm demonstrate the effectiveness of the proposed method. Particularly, the proposed approach achieves state-of-the-art performance in CALVIN (ABC→D) with only a single observation.

A limitation of the Diffusion Transformer Policy is that it requires multiple denoising steps during inference, which will impede the inference speed in the real application. In this paper, we focus on the modeling of complex and diverse actions. We think it is possible to improve the finetuning strategy with a few denoising steps to accelerate the inference speed. Meanwhile, the failure cases indicate it is important for the model to plan a right trajectory and grasp pose for picking object.

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
