# OpenReview forum: "Diffusion Transformer Policy"
_ICLR.cc/2025/Conference — ICLR 2025 Conference Withdrawn Submission_

### Official Review · Reviewer_3gSS · 2024-10-30

**Soundness:** 3
**Presentation:** 3
**Contribution:** 2
**Rating:** 6
**Confidence:** 4

**Summary:**

This paper identifies the challenges posed by large-scale robotic datasets such as the Open X-Embodiment Dataset, where variations in camera perspectives and changes across different robot embodiments create an enormous action space that traditional MLP action heads struggle to capture, leading to limited generalization capabilities. To address this, the authors introduce a causal transformer as the backbone of their conditional diffusion model. This model denoises action blocks using both language instructions and visual observations as input in an end-to-end fashion. This approach maintains scalability across large, cross-embodiment datasets while achieving superior generalization performance in new camera perspectives and environments.

**Strengths:**

The paper shows remarkable generalization capabilities across different simulated and real-world environments. Utilizing a large-scale Transformer model for action denoising demonstrates superior scalability compared to previous small-scale MLP-based denoising models and discretization actions. It shows significant advantages over existing methods, even when facing changes in camera perspectives or environmental variations.

**Weaknesses:**

I greatly respect the experimental design and the impressive generalization results presented in this paper. This work offers meaningful contributions on the engineering front. However, I have some questions regarding the academic novelty of the research.

Regarding the entire pipeline, employing the casual Transformer to accept both text tokens and image tokens as multimodal inputs is an approach that has been extensively used in prior research. Moreover, this paper's encoding of natural language and image features directly relies on existing models (e.g., CLIP and DINOv2) without showing any distinct insights or contributions from this study.

The core idea of this work is to use the Transformer as an end-effector to replace the conventional MLP head to capture complex action spaces and achieve higher generalization capabilities. However, the advantages of the Transformer architecture in capturing complex action spaces have already been extensively explored in the "Diffusion Policy: Visuomotor Policy Learning via Action Diffusion" paper. How does introducing the Transformer to improve scalability in this work differ from previous approaches?

In summary, this work significantly contributes to engineering and its applications within the related field. However, its academic innovation, particularly regarding theoretical contributions and architectural use, is not particularly strong.

**Questions:**

Can the authors clarify how their transformer-based approach differs from and improves upon previous work like the 'Diffusion Policy' paper and the "transformer" papers?
Especially in terms of architectural innovation?

---

### Official Review · Reviewer_bXvR · 2024-11-01

**Soundness:** 2
**Presentation:** 2
**Contribution:** 2
**Rating:** 3
**Confidence:** 4

**Summary:**

This paper presents Diffusion Transformer Policy,  which directly utilizes the larger transformers as a denoising network to denoise the continuous actions conditioned on language instruction and image observations.

The proposed model pre-trained on the OpenX-Embodiment Dataset achieves better generalization performance.

The authors validate the effectiveness of diffusion modeling and the larger transformer policy in two simulation environments.

**Strengths:**

The writing and organization of this paper are good.

The simulation experiments on Calvin and Maniskill are solid.

**Weaknesses:**

The fact that conducting pre-training on the Open-X dataset is a significant advantage of this paper. However, the real-world evaluation is too simple, only the  picking skill. This is not sufficient to prove the effectiveness of the methods in this paper compared to OpenVLA and OCto. If the evaluation scenarios include additional skills, it would be better.

It lacks the parameter scaling experiments of the Causal Transformer part. For example, the impact of different amounts of parameters and pre-training data on the results.

Why is it necessary to perform action decoding in a causal transformer with shared parameters? What if we simply use a larger DiT to replace the diffusion action head on the basis of Octo (such as [1] ) ? One advantage of doing so is that there is no need to pad the action vector. This paper lacks these experiments. I want to know which of these two design paradigms is better.

From the perspective of contribution to the community, this paper does not propose a powerful foundation model; at least the real-world experiments do not support this claim. On the other hand, the technology presented in this paper does not seem to be innovative; the initial diffusion policy could also utilize a transformer version.

[1] RDT-1B: a Diffusion Foundation Model for Bimanual Manipulation.

**Questions:**

Please see the weaknesses section.

---

### Official Review · Reviewer_5354 · 2024-11-01

**Soundness:** 2
**Presentation:** 2
**Contribution:** 2
**Rating:** 3
**Confidence:** 5

**Summary:**

This paper presents Diffusion Transformer Policy, a DiT-based generalist diffusion policy. Different from previous diffusion policies, DiT policy directly utilizes a causal transformer to perform diffusion denoising conditioned via in-context multimodal tokens. Specifically, language instruction and image observations are tokenized by CLIP and DINOv2, respectively. Then, the image tokens are compressed with Q-Former, conditioned by the language tokens via FiLM. After that, language tokens, compressed image tokens, diffusion timestep and the noisy actions are passed to a causal transformer for action denoising. Experiments in simulation environments (ManiSkill2, CALVIN) and real-world robot platform (Franka robot) demonstrate the performance of the proposed policy.

**Strengths:**

1. The paper utilizes DiT in an in-context conditioning style for continuous action chunk denoising, incorporating the strong scaling ability of Transformers for generalist robot policy learning. While several concurrent works also uses DiT to improve the manipulation policy, the idea has its novelty. The policy architecture is also well explained and illustrated.
2. Thorough ablations on trajectory length, observation length and execution steps.

**Weaknesses:**

1. **Real-World Experiments**.
    - **Task Setting**. The task settings are too easy with only “picking” operations. Despite that the authors claim that it is a challenging setup with small object (L370-371), I believe that only picking small objects would not be so difficult. I would suggest adding real-world manipulation experiments besides picking (and pick-and-place), *e.g.*, open drawer/door, pouring (with rotation actions), and long-horizon tasks, etc.
    - **Few-Shot Setting**. Collecting 50 trajectories for each task in the first three tasks (L368-369) results in totally 150 demonstrations, which is not a few-shot settings. Previous works [1, 2] has proved that 50 demonstrations in the real world are sufficient for the policy to learn complex behaviors far beyond picking, even w/o pre-training. Generally, using less than 10 demonstrations can be considered as “few-shot” [3, 4] for robot learning.
    - **Few-Shot Baselines**. While the authors employ Octo [5] and OpenVLA [6] as zero-shot generalization baselines, it would be consistent and convincing to have both methods also as few-shot generalization baselines.
    - **Few-Shot Results**. The current “few-shot fine-tuning” results in Table 4 seems a little low for the simple picking operations. Please provide additional object informations (Question 4 & 5) for task difficulty evaluations. The videos in the supplementary results lack unseen object evaluations (task-4 and task-5) and baseline performance.

2. **Improper Baseline Choices in CALVIN Experiment**. Developing an expressive generalist policy with a better generalization performance seems to be the key motivation of the paper. The generalization ability of the DiT policy is heavily boosted by large-scale OXE [7] pre-training (in CALVIN experiment, the policy outperform its w/o pre-training variant by 1.2 in success sequence length, L022-023), which is reasonable since large-scale pre-training exposes the policy to more robot data. Therefore, the baselines in CALVIN experiment should also access to a similar amount of robot data. Specifically,
    - **(a)** If the authors want to emphasize on the generalization ability of the DiT policy itself (not boosted by the large-scale pre-training), they should compare the DiT policy w/o pre-training  with some of the following policies: 3D diffusor actors [8], MDT [9], MT-ACT [10], Multitask-DP [11], etc., all trained with similar amount of data.
    - **(b)** If the authors want to emphasize that the DiT policy is suitable for large-scale pre-training and able to gain better generalization performance after large-scale pre-training, they should compare the DiT policy w/ pre-training with some of the following policies: RT-1/2-X [7], Octo [5], OpenVLA [6], MT-ACT [10] w/ OXE pre-training, Multitask-DP [11] w/ OXE pre-training, all fine-tuning with the same amount of data.

    Also, it is worth noticing that MDT [9] performs well in CALVIN benchmark even w/o large-scale robot data pre-training, though it has not been evaluated under the ABC→D settings in its paper. Evaluating MDT under this setting can serve as a strong baseline in CALVIN benchmark, even for situation (b). For most image-based policies [5, 6, 7, 9, 10, 11], the observation modalities can be easily aligned with the DiT policy (single-view RGB images) for fairness considerations.

3. **Writings**. There are several typos and unclear statement in the paper:
    - L137: QFormer is not properly cited for the first appearance.
    - L276: Table I, Disc ActionHead, SingleYCB “22.%”. Missing number.
    - L311: “…, and 1 dimension for gripper pose.” Is it “gripper position”?
    - L358: the first letter of “setup” should be capital.
    - L361: “… different from the scenes in Open X-Embodiment (Stone et al.)” Incorrect citation.

[1] Tony Z. Zhao, et al., “Learning Fine-Grained Bimanual Manipulation with Low-Cost Hardware”, RSS 2023.

[2] Cheng Chi, et al., “Diffusion Policy: Visuomotor Policy Learning via Action Diffusion”, RSS 2023.

[3] Vivek Myers, et al., “Policy Adaptation via Language Optimization: Decomposing Tasks for Few-Shot Imitation”, CoRL 2024.

[4] Kourosh Hakhamaneshi, et al., “Hierarchical Few-Shot Imitation with Skill Transition Models”, ICLR 2022.

[5] Octo Model Team et al., “Octo: An Open-Source Generalist Robot Policy”, RSS 2024.

[6] Moo Jin Kim, et al., “OpenVLA: An Open-Source Vision-Language-Action Model”, arXiv 2024.

[7] Open X-Embodiment Collaboration, et al., “Open X-Embodiment: Robotic Learning Datasets and RT-X Models”, ICRA 2024.

[8] Tsung-Wei Ke, “3D Diffuser Actor: Policy Diffusion with 3D Scene Representations”, CoRL 2024.

[9] Moritz Reuss et al., “Multimodal Diffusion Transformer: Learning Versatile Behavior from Multimodal Goals”, RSS 2024.

[10] Homanga Bharadhwaj, et al., “RoboAgent: Generalization and Efficiency in Robot Manipulation via Semantic Augmentations and Action Chunking”, ICRA 2024.

[11] Huy Ha, et al., “Scaling Up and Distilling Down: Language-Guided Robot Skill Acquisition”, CoRL 2023.

**Questions:**

1. For Maniskill2 experiment, are all policies OXE pre-trained or not? Please specify the experiment settings clearly.
2. MDT [3] also explores the use of multimodal-conditioned diffusion transformer model. Please clarify the difference between MDT and DiT policy.
3. In the real-world experiments, why is all the network fine-tuned with one observation and one step prediction (L375-376)? Especially for one-step prediction, since it has been proved that action chunking [8] (i.e., multi-step action prediction) is an effective technique for robotic manipulation policies [6, 7, 8, 9, 10], and the trajectory length ablation (L418-420) also supports the argument.
4. For the real world experiments, “it is a challenging setup with small object (L370-371)”. How small are the objects?
5. For the real world experiments, how many evaluation rollouts are conducted for zero-shot generalization and few-shot generalization, especially for the few-shot generalization experiments? What is the success metric, does the policy need to perform successful grasps at the first time, or multiple trials for the one rollout is allowed?

---

### Official Review · Reviewer_eFtp · 2024-11-02

**Soundness:** 2
**Presentation:** 2
**Contribution:** 2
**Rating:** 5
**Confidence:** 4

**Summary:**

This paper presents a different architecture / training paradigm for learning generalist robot policies from large scale data that uses a Causal Transformer model that denoises actions using a diffusion model training objective. Unlike other robot VLAs, their approach treats the entire model to denoise actions, while prior work only denoises with a small MLP layer. The authors present results in simulation environments, Maniskill 2 and Calvin, as well as real-world results on the Franka arm after pretraining on the Open X-Embodiment dataset.

**Strengths:**

- Extensive experimental results provided in several simulation environments and real-world task
- Reasonable set of baselines for comparison
- Reports state-of-the-art performance on the Calvin benchmark
- Clearly states training hyperparameters and experimental design to replicate their experiments

**Weaknesses:**

- Several typos and sentence style issues throughout the paper
- Motivation for the architecture modification is not clearly stated, not clear from reading the introduction.
- Figure 2 is easy to follow, but visually poorly constructed.
- Action tokenization procedure is unclear. Are the continuous actions being converted into discrete bins similar to some of the prior works? If it is not, then what is action tokenization is Section 3.1?
- Some architectural design choices are not motivated, i.e. what is the purpose of the Q-Former?
- If the issue is scalability, what happens if you increase the size of the MLP in the diffusion action head setup? 3-layer MLP seems small. Making the MLP head larger could provide a more fair comparison.
- Improvement in Table 1 seems to come from the SingleYCB case while the improvements in other tasks are modest and even worse for the PickCube task.
- The zero success rate performance seems questionable for OpenVLA and Octo in Table 3.

**Questions:**

- What is the point of using the Q-former?
- What happens if the diffusion action head is larger? It is not exactly clear what the main benefit of using the full causal Transformer as the diffusing model if the two approaches have roughly the same parameter count.

---

### Note · Authors · 2024-11-15

**Comment:**

Thanks for the comments from the reviewer.

**Withdrawal Confirmation:**

I have read and agree with the venue's withdrawal policy on behalf of myself and my co-authors.